# Object segmentation from *common fate*: Motion energy processing enables human-like zero-shot generalization to random dot stimuli

**Matthias Tangemann**        **Matthias Kümmerer**        **Matthias Bethge**

University of Tübingen, Tübingen AI Center
`matthias.{lastname}@bethgelab.org`

## Abstract

Humans excel at detecting and segmenting moving objects according to the *Gestalt* principle of "common fate". Remarkably, previous works have shown that human perception generalizes this principle in a zero-shot fashion to unseen textures or random dots. In this work, we seek to better understand the computational basis for this capability by evaluating a broad range of optical flow models and a neuroscience inspired motion energy model for zero-shot figure-ground segmentation of random dot stimuli. Specifically, we use the extensively validated motion energy model proposed by Simoncelli and Heeger in 1998 which is fitted to neural recordings in cortex area MT. We find that a cross section of 40 deep optical flow models trained on different datasets struggle to estimate motion patterns in random dot videos, resulting in poor figure-ground segmentation performance. Conversely, the neuroscience-inspired model significantly outperforms all optical flow models on this task. For a direct comparison to human perception, we conduct a psychophysical study using a shape identification task as a proxy to measure human segmentation performance. All state-of-the-art optical flow models fall short of human performance, but only the motion energy model matches human capability. This neuroscience-inspired model successfully addresses the lack of human-like zero-shot generalization to random dot stimuli in current computer vision models, and thus establishes a compelling link between the Gestalt psychology of human object perception and cortical motion processing in the brain.
Code, models and datasets are available at `https://github.com/mtangemann/motion_energy_segmentation`.

## 1   Introduction

Motion is a powerful cue that humans use to detect and segment visual objects. A striking example are camouflaged animals, which are difficult to spot when stationary but become much easier to detect when moving. Motion segmentation in humans is believed to be driven by the principle of common fate [47, 48, 42], which posits that elements that move together, belong together. Remarkably, human perception generalizes this principle in a zero-shot fashion to novel textures or moving random dots. For example, the seminal work by Johansson [19] showed that humans can easily detect biological motion from only few moving dots. More recently, Robert et al. [31] introduced random dot stimuli called *object kinematograms* that preserve the motion in a video while ensuring that static appearance cues are uninformative about the video contents (Fig. 1, example video in the supplemental material).

38th Conference on Neural Information Processing Systems (NeurIPS 2024).

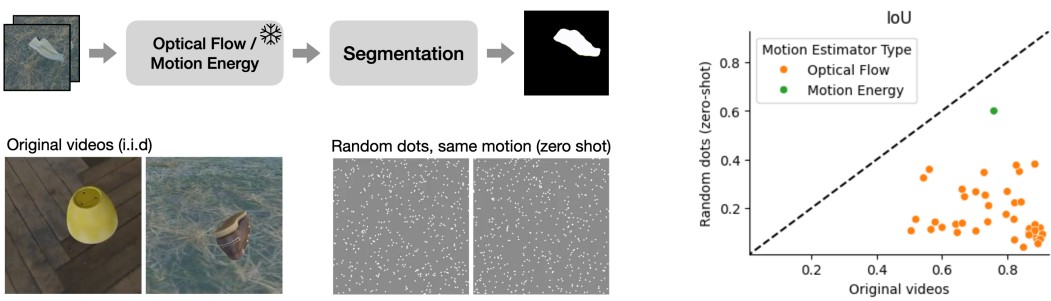

Figure 1: We compare state-of-the-art optical flow estimators and a neuroscience inspired motion energy model on a figure-ground segmentation task. For evaluation, we use random dot stimuli with the same motion patterns as the original videos, but for which the appearance of each individual frame is informative (example video in the supplemental material). The neuroscience inspired model generalizes to these stimuli much better than state-of-the-art optical flow models.

Nevertheless, humans were able to classify the animals and objects in these videos based on motion information alone.

In this work, we seek to understand the computational basis for appearance-agnostic motion perception in humans which enables this zero-shot generalization to random dots. Recent advancements in computer vision models for motion segmentation enable accurate segmentation of moving objects in natural videos based on a combination of optical flow estimation networks with downstream segmentation networks (e.g., [51]). However, it remains untested whether these models generalize in a similar way as human perception. Since the motion estimation stage is critical for segmenting moving objects, we focus on testing a broad range of state-of-the-art optical flow methods in combination with a fixed segmentation network. Our analysis reveals that existing computer vision approaches do not generalize in a human-like manner: Many high-performing models on natural stimuli perform near chance level for random dots.

In the primate visual cortex, area MT is known to be involved in motion perception and interpretation. Computational models for this area based on *motion energy* were proposed almost forty years ago [1, 46] and since then have been shown to predict key characteristics of neural firing patterns [37]. Instead of matching deep features between two frames, these models rely on spatio-temporal filtering in pixel space combined with a post-processing stage to resolve ambiguities. We demonstrate that this mechanism can be successfully integrated with deep neural networks for motion segmentation in realistic videos and reaches the performance of early deep learning based optical flow models on the original, textured videos—which is remarkable considering that the motion energy model was developed to explain the tuning of individual neurons and has several orders of magnitude fewer parameters than typical optical flow networks. Crucially, the motion energy model substantially outperforms all tested optical flow models in zero-shot generalization to moving random dots. In a direct comparison with humans in a controlled psychophysics study, the motion energy based approach is the only model that can match human capability.

In summary, our paper makes the following contributions:

- We show that a a broad range of state-of-the-art optical flow methods do not support human-like motion segmentation that generalizes to random dot patterns.
- We demonstrate that a classical neuroscience model can be successfully integrated with deep neural networks and generalizes to random dot stimuli.
- We conduct a psychophysical experiment to directly compare random dot motion segmentation in humans and machines. While state-of-the-art optical flow models fall short of human performance, the motion energy model can match it.

These results establish a compelling link between the Gestalt psychology of human object perception and cortical motion processing in the brain, showing that a motion energy approach can overcome the lack of human-like zero-shot generalization to random dot stimuli in current computer vision models. Integrating this mechanism with state-of-the-art optical flow methods is promising path towards more robust motion estimation models.

## 2  Related Work

**Motion energy.**    Modelling motion perception in humans has been frequently approached using motion energy models. These models exploit the fact that a moving pattern corresponds to oriented edges when considering a video as a spatio-temporal volume [1, 46]. Several models have been proposed that build on this principle, aiming to explain the tuning properties of neurons found in visual areas V1 and MT [37, 15, 32, 28]. With few exceptions [41, 38], these models have not been used as a motion estimation models in a computer vision context. Our work is the first to study motion energy models for moving object segmentation.

**Optical flow estimation.**    Optical flow traditionally has been formulated as an optimization problem with the goal of finding good matches between two frames [16]. During recent years, optimization based methods have been superseded by deep neural networks that frame optical flow estimation as an end-to-end regression task. FlowNet [11] pioneered this approach with a CNN that optionally includes an explicit temporal matching operation. Following works contributed better training data and proposed coarse-to-fine architectures to predict optical flow [17, 39, 40] which lead to substantial performance improvements. More recently, models that iteratively refine a high resolution optical flow map [43, 18] and Transformer-based models [36, 53, 54] have further improved state-of-the-art. Some works have compared optical flow models to human motion perception [56, 41], however not in the context of motion segmentation.

**Motion Segmentation.**    The typical approach to motion segmentation is using optical flow as input for a downstream segmentation model. One classic line of work computes point trajectories from optical flow and then clusters the trajectories to segment moving regions [6, 29, 20]. Classical geometric approaches to motion segmentation have been combined with deep learning in later work [4, 5]. More recently, purely deep learning based approaches have been able to improve state-of-the-art [44, 9, 22, 23, 51]. To achieve high performance on classical motion segmentation datasets, the optical flow based motion segmentation is typically combined with appearance based segmentation [9, 52]. In this work, we evaluate generalization to random dot stimuli for which appearance is not informative, so we focus on purely motion-driven approaches.

## 3  Methods

The aim of this work is to evaluate which computational models match the capabilities of humans for zero-shot motion segmentation of random dot patterns. We follow the standard motion segmentation approach in computer vision and first use a motion model to estimate the motion in an input video, followed by a segmentation network that predicts the foreground mask. In order for models to perform well on zero-shot segmentation of random dot patterns, it is critical that the motion estimator used by the model generalizes well to these random dot stimuli. Ideally, the motion estimator would be invariant to changes in texture. Therefore, we focus on the motion estimation stage by evaluating a broad range of optical flow models in comparison to a neuroscience inspired motion energy model. As a segmentation model, we use the same segmentation architecture for all motion estimators which we train from scratch for every model.

### 3.1  Optical Flow Models

We use a range of optical flow models that includes all major deep learning based approaches to optical flow estimation. FlowNet 2.0 [17] was the first CNN based model that reached the performance of classical, optimization based methods. We consider three variants of the model using different combinations of subnetworks. PWC-Net [39] introduced a multi-scale approach that combined operations from classical approaches (such as cost volumes and warping), with components from deep learning. Different from previous models, RAFT [43] is not based on a coarse-to-fine approach but rather on iterative refinement of a high resolution optical flow map derived from multi-scale correspondences. GMA [18] extends the RAFT architecture by introducing a Transformer-based module to better handle occlusions, which have been shown to be difficult for previous models. More recently, GMFlow [53, 54] and FlowFormer++ [36] have been proposed as fully Transformer-based architectures for optical flow estimation.

We use the implementations and checkpoints of these models from the MMFlow library [8], except for FlowFormer++ and GMFlow for which we use the implementations and checkpoints provided by the respective authors[1][2]. For each architecture, we consider checkpoints trained on different datasets that are common in the field, including the FlyingChairs [11], FlyingThings3d [24], Sintel [7] and KITTI [25, 26]. In total, we evaluate 40 optical flow models.

We apply the models to predict multi-scale optical flow, in order to match the multi-scale features predicted by the motion energy model. All of the optical flow models internally use several scales to predict optical flow. However, this representation is followed by non-trivial processing to combine motion information across scales, so that using this internal representation directly would most likely lead to inferior performance. Therefore we use the unmodified models and scale the final optical flow prediction to the desired resolutions using bilinear interpolation.

## 3.2 Motion Energy Model

Motion energy models are based on the insight that a motion pattern in a video corresponds to a spatio-temporal orientation when the video is considered as an $x$-$y$-$t$ volume [1, 46]. The motion at every pixel can therefore be estimated by using spatiotemporal filters that respond to a particular motion direction and speed. This mechanism has important differences from the optical flow models discussed before. All of the optical flow models compute deep features for two frames individually and match these features between two frames to estimate motion. The spatio-temporal filters in motion energy models on the other hand operate directly in pixel space. This approach leads to more ambiguous matches, which are typically resolved by considering more than two frames, and a postprocessing stage.

In this study, we build on the influential motion energy model by Simoncelli & Heeger [37]. In addition to the oriented filters described above, this model introduced a second stage that implements an *intersection of constraints* construction [13, 2] in order to resolve ambiguities of the linear filter responses. This motion energy model can be implemented as a CNN with the architecture shown in Figure 2. We derived the weights of the CNN from the the parameters of the original model and verified that our PyTorch [30] implementation of the motion energy model equals the original MATLAB implementation[3] up to numerical differences. Following the original model, we apply the model for five different input scales that are obtained by repeatedly blurring and downsampling the input by a factor of two. To streamline the implementation, we do not scale the activations after every layer and experimentally verified that this change does not affect downstream performance for motion segmentation.

## 3.3 Segmention model

We use a coarse-to-fine segmentation network to predict per-pixel logits for the respective pixel belonging to the foreground object (Figure 2). Input to the segmentation model are the multi-scale motion energy maps or multi-scale optical flow maps as predicted by the models described earlier. At each scale, the segmentation model consists of three components: The *input projection* layer predicts motion features for each scale. The core of the network is a *refinement CNN* that aggregates features across scales. At each scale, the refinement CNN concatenates the motion features from the current scale with the refined representation from all previous scales and predicts the refined representation for the current scale. Finally, the *output projection* layer predicts the segmentation given the refined representation from the finest scale. All layers except for the output projection are followed by a CELU nonlinearity [3] and instance normalization [45]. The parameters of the components are shared across the stages, so that the network is essentially a recurrent neural network that integrates information from coarsest to the finest scale in order to predict a segmentation.

## 3.4 Training

All models are trained on a synthetic video dataset that we generated using the Kubric library[14]. Each video shows a single moving object in front of a moving background. The 3D objects and

---

[1]`https://github.com/XiaoyuShi97/FlowFormerPlusPlus`
[2]`https://github.com/autonomousvision/unimatch`
[3]`https://www.cns.nyu.edu/~lcv/MTmodel/`

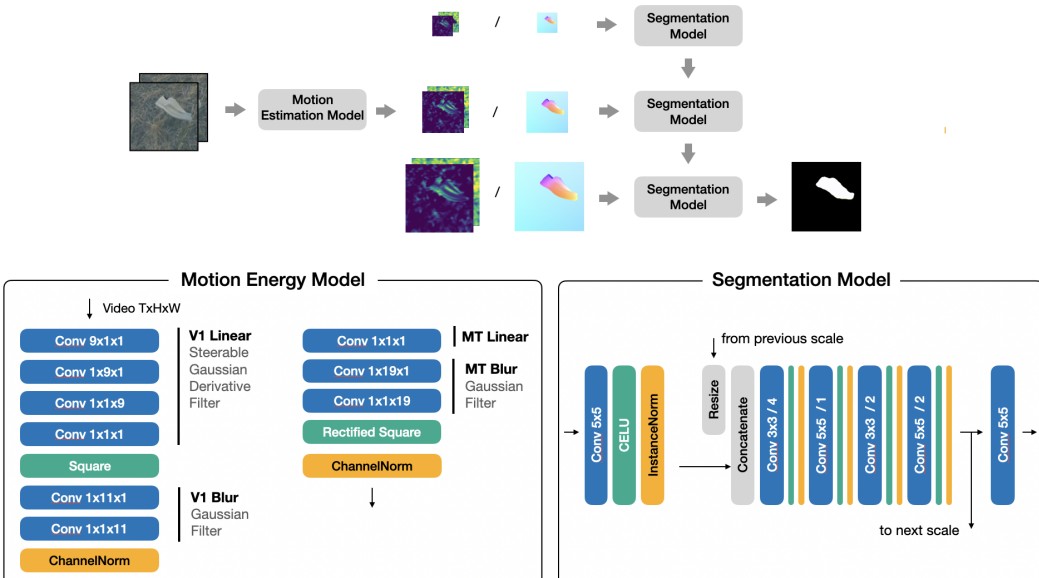

Figure 2: (*top*) Our motion segmentation architecture: The motion estimation predicts multi-scale optical flow or motion energy, the segmentation model predicts the moving foreground region. (*bottom left*) The motion energy model is implemented as a CNN. The weights are chosen such that the CNN is equivalent to the original model by [37]. (*bottom right*) The segmentation model combines motion features across scale and predicts a binary segmentation at the input resolution.

backgrounds used for dataset generation are scans of everyday objects and scenes, resulting in highly realistic renderings. We used 901 videos for training and 100 test videos, each having 90 frames at 30Hz. The training and test videos used different sets of object and backgrounds but are otherwise sampled from the same distribution. The code and hyperparameters for generating videos, as well as the rendered dataset, are publicly available[4].

For all models, we freeze the weights of the motion estimator and only train the downstream segmentation network. As common for binary motion segmentation, we use per pixel binary cross entropy to the ground truth masks as loss. We use the Adam optimizer [21] with a learning rate of $1e - 4$ for all models and train for 40.000 steps using a batch size of 8. All models are trained on NVIDIA GeForce RTX 2080 Ti GPUs with 12GB of VRAM. Depending on the computational requirements of the motion model, training the segmentation model on a single GPU takes between 2 and 6 hours.

### 3.5 Zero-shot evaluation on random dot stimuli

We evaluate models on the original test videos as well as random dot stimuli generated for all test videos based on the ground truth optical flow. We use the same procedure as [31] for generating random dot stimuli using 500 dots with a lifetime of 8 frames, which matches the dot density and lifetimes.

We apply all models using a shifting window approach for the full length videos, but excluding the first and last four frames so that the window is fully contained within the video for all models. For evaluation, we obtain a binary prediction by thresholding with 0.5 and measure performance by computing IoU and F-Score for each frame individually and then averaging over the test set.

---

[4]https://github.com/mtangemann/motion_energy_segmentation

# 4 Results

## 4.1 Zero-Shot Random Dot Segmentation

Table 1 summarizes the motion segmentation performances achieved when using different motion estimators, both on the i.i.d. test set and the corresponding random dot stimuli. For a better overview, we visualize the performances as measured by IoU in Figure 1 and in the appendix.

| Motion Estimator | Training Dataset | Original IoU | Original F-Score | Random Dots IoU | Random Dots F-Score |
|---|---|---|---|---|---|
| Motion Energy (ours) | - | 0.759 | 0.845 | **0.600** | **0.718** |
| GMFlow | Flying Things 3D | 0.885 | 0.925 | 0.381 | 0.493 |
| (2 scales, 6 refinements) | Sintel | 0.823 | 0.874 | 0.222 | 0.315 |
| | Mixed | 0.823 | 0.874 | 0.069 | 0.106 |
| FlowNet2 SD | FlyingChairs | 0.828 | 0.884 | 0.377 | 0.499 |
| FlowNet2 CSS | FlyingThings3D | 0.837 | 0.896 | 0.351 | 0.469 |
| | FlyingChairs | 0.735 | 0.818 | 0.252 | 0.359 |
| PWC-Net | FlyingThings3D | 0.729 | 0.815 | 0.347 | 0.469 |
| | FlyingChairs | 0.544 | 0.662 | 0.324 | 0.442 |
| | KITTI | 0.600 | 0.715 | 0.121 | 0.193 |
| FlowNet2 | FlyingChairs | 0.662 | 0.757 | 0.278 | 0.380 |
| | FlyingThings3D | 0.821 | 0.877 | 0.154 | 0.241 |
| FlowNet2 CS | FlyingThings3D | 0.800 | 0.867 | 0.269 | 0.374 |
| | FlyingChairs | 0.669 | 0.760 | 0.247 | 0.359 |
| GMFlow (2 scales) | Flying Things 3D | 0.704 | 0.793 | 0.267 | 0.373 |
| | Sintel | 0.733 | 0.812 | 0.253 | 0.351 |
| | Mixed | 0.743 | 0.822 | 0.210 | 0.300 |
| GMFlow (1 scale) | Mixed | 0.843 | 0.897 | 0.225 | 0.308 |
| | Flying Things 3D | 0.797 | 0.859 | 0.174 | 0.244 |
| GMA (+P) | KITTI | 0.520 | 0.630 | 0.154 | 0.224 |
| | FlyingChairs | 0.646 | 0.717 | 0.101 | 0.142 |
| | Mixed | 0.895 | 0.932 | 0.054 | 0.078 |
| | FlyingThings3D | 0.850 | 0.894 | 0.039 | 0.057 |
| FlowFormer++ | Flying Chairs | 0.741 | 0.800 | 0.144 | 0.218 |
| | Flying Things 3D | 0.901 | 0.935 | 0.119 | 0.182 |
| | Sintel | **0.908** | **0.942** | 0.092 | 0.140 |
| | Flying Things 3D | 0.902 | 0.938 | 0.072 | 0.113 |
| GMA | KITTI | 0.579 | 0.679 | 0.143 | 0.214 |
| | FlyingChairs | 0.643 | 0.724 | 0.134 | 0.194 |
| | FlyingThings3D | 0.867 | 0.909 | 0.100 | 0.150 |
| | FlyingThings3D + Sintel | 0.890 | 0.928 | 0.089 | 0.132 |
| | Mixed | 0.890 | 0.927 | 0.077 | 0.109 |
| GMA (P-only) | FlyingChairs | 0.663 | 0.743 | 0.138 | 0.207 |
| | KITTI | 0.567 | 0.650 | 0.112 | 0.159 |
| | Mixed | 0.881 | 0.920 | 0.093 | 0.141 |
| | FlyingThings3D | 0.867 | 0.909 | 0.090 | 0.142 |
| RAFT | FlyingThings3D + Sintel | 0.886 | 0.924 | 0.132 | 0.180 |
| | FlyingThings3D | 0.869 | 0.909 | 0.116 | 0.172 |
| | Mixed | 0.885 | 0.925 | 0.108 | 0.145 |
| | KITTI | 0.506 | 0.600 | 0.107 | 0.147 |
| | FlyingChairs | 0.647 | 0.718 | 0.100 | 0.147 |

Table 1: Model performances for the i.i.d. test videos and zero shot to the corresponding random dot stimuli with the same motion patterns. For all motion estimators, the same segmentation network is used to predict the figure-ground segmentation. Results are grouped by the motion estimation model and ordered by the performance on the random dot stimuli.

**Recent optical flow methods perform strongly on the original videos**. FlowFormer++ works best on our dataset with an IoU of 90.8%, closely followed by a GMA variant that reaches 89.5% IoU.

These results parallel the strong performance of recent Transformer-based architectures on standard optical flow benchmarks. The motion energy based model only achieves a performance of 75.9% IoU and lags behind state-of-the-art optical flow models, but performs similar as earlier deep learning based optical flow models. This result is remarkable when considering that the motion energy model predates the deep learning models by several decades and has not been tuned for dense, end-to-end motion prediction. Within each model, the checkpoints from the FlyingThings3d dataset tend to perform best for the original videos. The FlyingThings3d dataset contains renderings of 3D objects undergoing rigid motion, so arguably it is the most similar dataset compared to the one used in this study.

**Motion energy generalizes much better to random dots**. The motion energy based model reaches an IoU of 60.0%, which outperforms the performance of the second best model by more than 20 percentage points. Strikingly, the FlowFormer++ and GMA models that performed best on the original videos generalize particularly bad to the random dot stimli (IoU < 10%). Overall, more dated optical flow architectures such as FlowNet2 variants and PWC-Net tend to generalize better to random dot stimuli than more recent approaches. An interesting exception is GMFlow, which reached an IoU of 38.1% and performed best among all optical flow models. We do not observe a clear effect of the training dataset.

We visualize model predictions in Figure 3. For the original videos, the quality of the predicted optical flow varies but allows for a clear segmentation of the moving object. The object is also clearly represented in the motion energy maps, with some feature maps responding highly to the background and others to the moving object. The motion energy maps however tend to be noisier than the optical flow predictions, which explains the lower performance of the motion energy model for the clean videos.

The random dot stimuli exhibits the same motion as the original video, so the prediction of an ideal motion estimator would be unchanged. The optical flow methods however fail to properly estimate the motion of the foreground object. While some methods like FlowNet 2.0 and PWC-Net predict a highly noisy motion pattern that roughly matches the location of the foreground object, many optical flow estimators fail to detect the foreground motion at all. The motion energy on the other hand looks highly similar for the random dot stimulus and the original video, allowing the motion energy segmentation to generalize well in this case.

## 4.2   Ablation study

As an ablation study, we evaluated whether the performance of the motion energy segmentation model can be improved by learning the parameters of the motion model. We tested different combinations of layers in the motion energy CNN that are fixed, finetuned or trained from scratch and trained them end-to-end with the segmentation model.

The results in table 2 show that the original weights of the model allow for the best generalization to random dots. This is remarkable when considering that the weights of the motion energy model have been originally selected to explain the tuning properties of individual neurons, but not for image-computable motion estimation. Some of the configurations however outperformed the original weights on the original videos. So while the network architecture allows for generalization in principle, all our models trained by gradient descent converged to solutions that performed well on the training data but did not generalize.

As a further ablation study, we removed or replaced layers of the motion energy model. The results in the supplemental information suggest that the pooling and normalization layers are particularly important for generalization to random dots. More details and further experiments are provided in the supplemental information.

## 5   Human Machine Comparison

The previous results have revealed differences between different motion estimation models in terms of generalization to random dots. While it is known that humans can recognize objects in random dot stimuli without prior training [31], the ability to segment objects in moving random dot patterns has not been quantified before. We therefore conduct a human subject study in order to directly compare the zero-shot generalization to random dots in humans and machines.

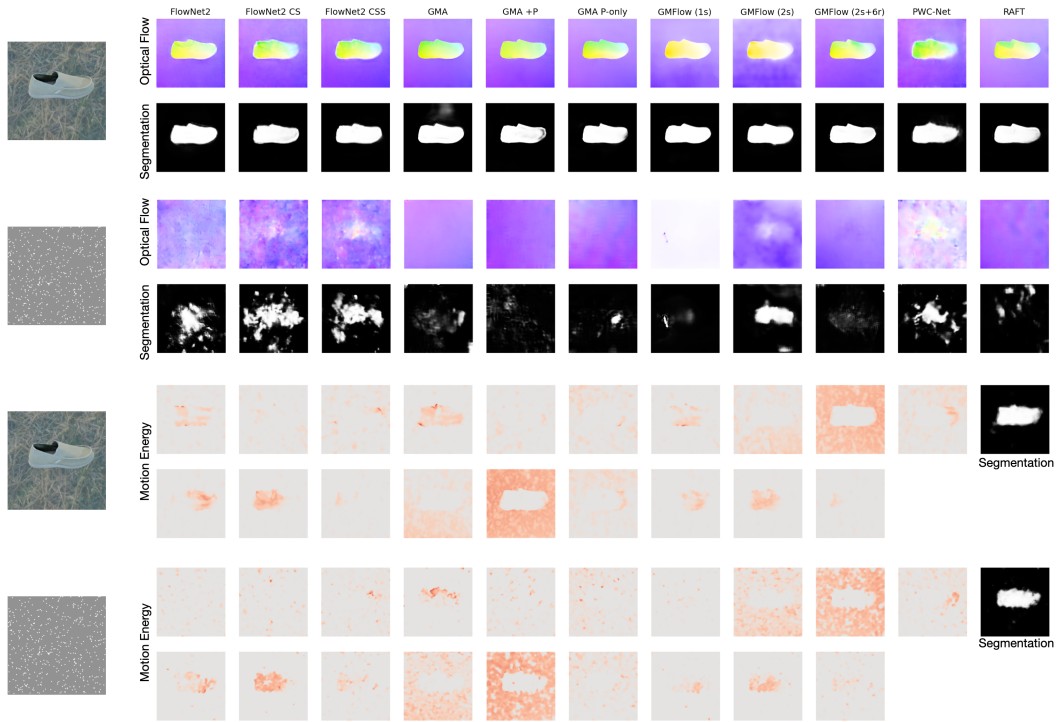

Figure 3: Example predictions for different motion estimators. The motion pattern in the random dot stimulus is the same as in the original video. While the optical flow estimates are highly accurate for the original videos, the models struggle with the random dot stimuli that exhibit the same motion. The activations of the motion energy model model however generalize well to the random dot stimuli, enabling to detect and segment the foreground object.

| V1 Linear | V1 Blur | MT Linear | MT Blur | Original | | Random Dots | |
|---|---|---|---|---|---|---|---|
| | | | | IoU | F-Score | IoU | F-Score |
| fix | fix | fix | fix | 0.759 | 0.845 | **0.600** | **0.718** |
| fix | fix | finetune | fix | 0.776 | 0.856 | 0.563 | 0.686 |
| finetune | fix | finetune | fix | 0.804 | 0.873 | 0.468 | 0.599 |
| fix | fix | scratch | scratch | 0.794 | 0.873 | 0.463 | 0.583 |
| finetune | fix | scratch | scratch | **0.827** | **0.887** | 0.395 | 0.508 |
| finetune | finetune | scratch | cratch | 0.600 | 0.717 | 0.162 | 0.246 |
| scratch | fix | scratch | fix | 0.660 | 0.752 | 0.052 | 0.087 |
| scratch | scratch | scratch | scratch | 0.593 | 0.702 | 0.027 | 0.048 |

Table 2: Comparison of using the original weights (fix), finetuning the original weights (finetung) or training from scratch (scratch) for the layers of the motion energy model.

Due to the inherent difficulty in directly evaluating the segmentation perceived by humans, we employed a shape identification task as a surrogate requiring segmentation (Figure 4). Each trial involved a random target shape and a distractor shape from the Infinite dSprites dataset [12]. The random dot stimulus shows the target shape moving linearly across the center of the image with random motion direction and speed. After the video concluded, participants were shown clean renderings of the target and distractor shapes and were required to select the shape that they perceived in the random dot stimulus. Since the shape alternatives were unknown while the random dot stimulus was shown, participants had to segment and memorize the shape in the random dot video and then compare it to the shape choices afterward. Therefore, performing well on this task necessitates sufficiently good segmentation of the moving shapes within the motion patterns of the random dot stimuli.

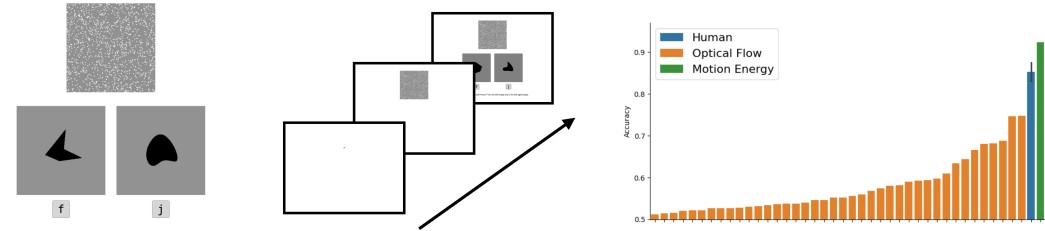

Figure 4: We compare humans and machines using a *random dot shape identification* task as a proxy to measure segmentation in humans. Shown a video of random dots, participants have to respond which of two shapes was perceived in the video. Humans outperformed all optical flow based models, but not the motion energy based model for this task. More details are provided in the supplemental material.

We performed the study in a controlled vision lab environment, where participants viewed the experiment on a VIEWPixx 3D LCD monitor (1920x1080, 120Hz) with the distance fixed to 65cm using a chin rest. The duration of all videos was 1s at a framerate of 30 Hz. Overall, we collected data from N=13 subjects, of which we excluded one subject due to insufficient visual acuity (remaining: N=12, 4 female, 8 male). Among the subjects where both trained vision scientists and naive subjects.

We evaluated all models on the same stimuli as human subjects. Given a random dot video, we applied the respective model to segment the video and selected the shape option that better matched the prediction as measured by IoU.

The results in Figure 4 show that all models based on optical flow are clearly outperformed by humans. Many of the optical flow based models perform near chance level, while some models reached a non-trivial performance. Overall, more recent optical flow models that perform very well on the original videos appear to generalize worse to this task, with GMFlow [53, 54] being a noteable exception. Different from the optical flow models, the motion energy based approach is the only model to match and even outperform human performance. More detailed results in the supplemental information show that the motion energy segmentation model performs on par with the highest performing participants of the study.

## 6   Limitations

To allow for comparing a large number of motion estimation models with a reasonable computational budget we made compromises for other modeling aspects. We limited the size of the segmentation network to allow for efficient training but performed a control experiment to show that using a more sophisticated segmentation network does not improve generalization (see supplemental information). Moreover, we used the same training schedule for all models but ensured that our setting supports all models adequately by visually inspecting the loss curves.

When comparing humans and machines we did not model several factors that are expected to influence human performance, such as the impact of internal noise and attentional lapses. As common in psychophysics experiments, several subjects reported making accidental errors for few examples [50] which negatively affects performance. So even a model that perfectly replicates the motion processing algorithm in humans is not expected to perfectly replicate human behavior in our setting.

## 7   Broader impact

This work is highly interdisciplinary, bridging state-of-the-art computer vision motion segmentation algorithms with the principles of Gestalt psychology and the neuroscience of cortical motion processing in the brain. By showing that a neuroscience-inspired motion energy model can outperform conventional optical flow models in zero-shot generalization to random dot stimuli, the study highlights the potential for integrating biological insights into AI systems. Benefits of broader impact include the development of more robust and human-like AI systems, educational value, and the creation of AI systems that are more aligned with human cognition.

# 8 Discussion

Computational models for motion estimation have a long history in both computational neuroscience and computer vision. Shallow models based on spatio-temporal filtering in pixel space have been able to predict neural activity in brain areas related to motion perception [37, 32] and are compatible with a range of phenomena in human perception [57]. In computer vision, models based on matching deep features between two frames have continuously improved performance over the last years and are successfully applied in a range of downstream tasks. Despite these successes, our study reveals a striking gap between deep optical flow networks and human perception: While humans generalize the common fate Gestalt principle to zero-shot segmentation of random moving dots, the optical flow models fail to generalize to these stimuli. Furthermore, we show that a classic motion energy approach can be scaled to realistic videos while matching human generalization capabilities.

The great success of deep neural networks in computer vision has spawned interest in using DNNs also as a model for human vision, in particular for core object recognition [55]. In the same spirit, deep neural networks might be promising models for human motion perception [56]. While promising, our study parallels findings for core object recognition that show striking differences between human perception and DNNs [49]. For motion perception, however, we show that is possible to combine classical models from computational neuroscience with the scalability of deep learning. Further integration of these modeling traditions is a promising path towards image computable models of human motion perception [41].

While closing the gap between human perception and machine vision is crucial for computational neuroscience, we believe that computer vision likely profits from better alignment with human vision as well. Humans still greatly outperform machines in terms of robustness and efficiency. Our study suggests a substantial entanglement of motion estimation with appearance in DNNs, which might also be linked to the lack of robustness observed in state-of-the-art motion estimators [33, 34]. Computational principles that better match human vision should be considered as promising candidates for addressing these issues.

Finally, we argue that deep learning based models as presented in our work have the potential to greatly improve our understanding of motion perception in humans. Low-level mechanisms for motion estimation and higher level processes for motion interpretation have been mostly studied in isolation [27]. In our work we follow a more holistic approach by studying the effects motion detection mechanisms on the perception of moving objects, which offers several unique opportunities. First, it is not necessary for most downstream tasks to perfectly estimate the physically correct motion. For example, segmenting moving objects does require precise information about object boundaries while other mistakes are less critical. Studying motion estimation and interpretation jointly allows to better understand viable compromises in estimation accuracy as the basis for more efficient processing. Second, studying end-to-end models of motion estimation and interpretation advances our understanding of how neural mechanisms give rise to behavior. DNNs are a particularly promising modeling approach positioned in a "Goldilocks zone" regarding the trade-off between biological plausibility and scalability to natural stimuli and tasks [10]. In this vein, our work establishes a compelling link between cortical mechanisms for motion estimation and the Gestalt psychology of human object perception.

In the future, this work can be extended in several directions. While scaling remarkably well, the original motion energy model is not able to match the performance of state-of-the-art optical flow methods on natural scenes. We see integrating principles from computational neuroscience with techniques from deep learning as a promising path towards closing this gap [41]. Moreover, training the parameters of the CNN implementation of the motion energy model jointly with the segmentation model did not lead to a generalizable solution. How humans learn generalizable motion perception from data, or to which degree this capability is innate, are important questions for future research. Finally, in the spirit of the neuroconnectionist research programme [10] we see our model as an executable hypothesis for motion perception in the human brain. While matching human performance in terms of generalization to moving random dots, this model might well fail to capture other aspects of human motion perception. Further evaluating and extending models of motion perception to capture a diverse range of phenomena is an exciting path towards a holistic understanding of human perception.

## Acknowledgments and Disclosure of Funding

This work was supported by the German Research Foundation (DFG): SFB 1233, Robust Vision: Inference Principles and Neural Mechanisms, TP 4, project number: 276693517. The authors thank the International Max Planck Research School for Intelligent Systems for supporting MT. We thank Felix Wichmann, Thomas Klein and all other members of the Wichmann-Lab for supporting and testing the human machine comparison study, and Larissa Höfling for valuable feedback on the manuscript.

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

# Supplemental information

## A   Additional details about the results

For an additional overview, view visualize the segmentation performances on random dot stimuli as reported in Table 1.

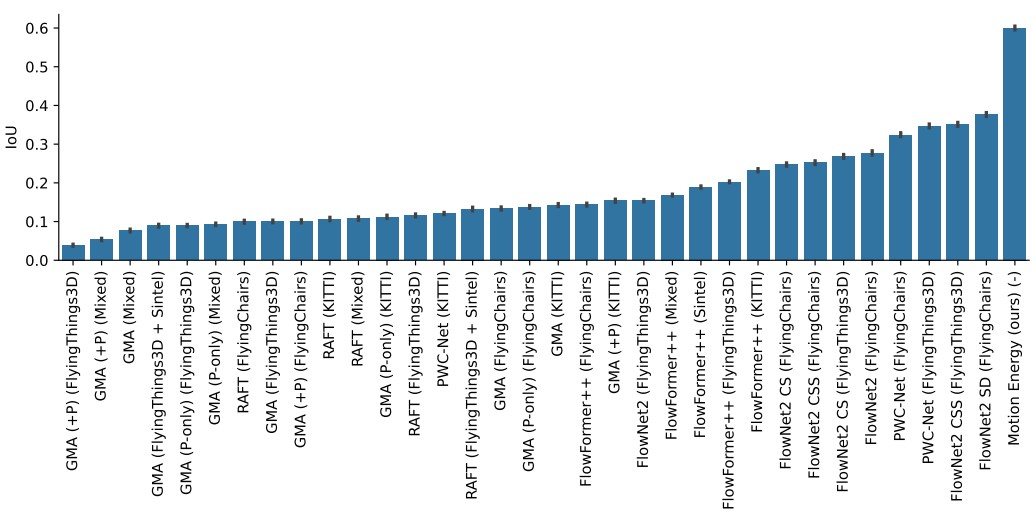

Figure 5: Segmentation performances of the evaluated models on the random dot stimuli. Same data as in Table 1.

## B   Additional experiments

### B.1   Importance of components of the motion energy model

We conducted an additional ablation study in order to better understand which aspects of the motion energy model are essential for generalization to random dot stimuli. We removed or replaced individual layers as described in Table 3 and trained the ablated models from scratch using in the same way as the baseline model.

The results in Table 2 hint at the normalization and pooling layers being important for generalization. When the Gaussian pooling layers are removed completely, the performance on original videos even slightly improves while the generalization to random dot stimuli is substantially reduced.

Replacing the squaring-based nonlinear layers with ReLU layers, however, hardly changes the model's performance.

| Condition | Original | | Random Dots | |
|---|---|---|---|---|
| | IoU ↑ | F-Score ↑ | IoU ↑ | F-Score ↑ |
| Baseline | 0.759 | 0.845 | 0.600 | 0.718 |
| Replace RectifiedSquare → ReLU (MT) | 0.753 | 0.838 | **0.609** | **0.725** |
| Replace Square → ReLU (V1) | 0.770 | 0.854 | 0.536 | 0.663 |
| Remove MT Linear | 0.768 | 0.856 | 0.481 | 0.609 |
| Remove MT | 0.770 | 0.854 | 0.451 | 0.583 |
| Remove Blur (V1, MT) | **0.801** | **0.872** | 0.421 | 0.540 |
| Replace ChannelNorm → InstanceNorm (V1, MT) | 0.592 | 0.703 | 0.230 | 0.340 |
| Remove Normalization (V1, MT) | 0.400 | 0.516 | 0.018 | 0.018 |

Table 3: Ablation study: Performance of the model on original videos and corresponding random dot stimuli with various layers of the motion energy model removed or replaced. Results are ordered by IoU on the random dot stimuli.

## B.2 Multi-frame optical flow

The motion energy model uses a window of 9 frames as input, while typical optical flow methods estimate correspondences between only two frames. To rule out the possibility that the results observed in our paper are mainly explained by the different input window lengths, we perform an ablation study in which we apply optical flow methods using the same 9 frame windows. For each window, we compute the optical flow between the central frame, for which the segmentation has to be predicted, to the 8 other frames in the window. The stacked optical flow fields are then used as the input to the segmentation network.

The results in Table 4 and Figure 6 show some improvement on the original videos but an ever wider gap to the motion energy model in terms of of generalization to random dots. The differences between the motion energy and optical flow models therefore cannot be explained by the different input lengths.

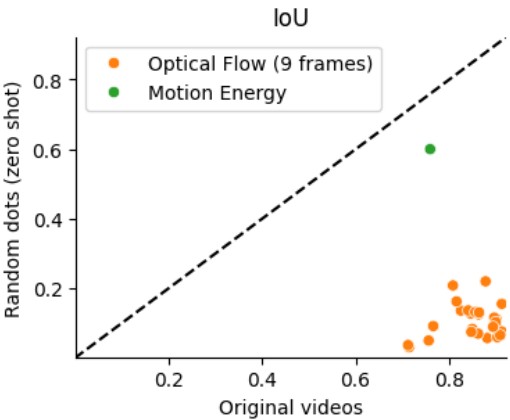

Figure 6: Performance of multi-frame optical flow based models on the original videos and corresponding random dot videos.

## B.3 Comparison with state-of-the-art motion segmentation

In our study we used a relatively small segmentation network downstream to the respective motion estimator. State-of-the-art motion segmentation models typically target multi-object segmentation in real world videos and therefore use more complex segmentation networks. In order to verify that the results in our paper are not caused by using a smaller segmentation network, we evaluated the state

| Motion Estimator | Training Dataset | Original | | Random Dots | |
|---|---|---|---|---|---|
| | | IoU | F-Score | IoU | F-Score |
| Motion Energy (ours) | - | 0.759 | 0.845 | **0.600** | **0.718** |
| FlowNet2 SD | FlyingChairs | 0.878 | 0.928 | 0.221 | 0.325 |
| FlowNet2 | FlyingChairs | 0.808 | 0.868 | 0.209 | 0.300 |
| | FlyingThings3D | 0.881 | 0.929 | 0.058 | 0.100 |
| PWC-Net | FlyingChairs | 0.816 | 0.886 | 0.163 | 0.250 |
| | FlyingThings3D | 0.825 | 0.886 | 0.137 | 0.221 |
| | KITTI | 0.712 | 0.811 | 0.038 | 0.060 |
| RAFT | FlyingThings3D + Sintel | **0.912** | **0.948** | 0.156 | 0.222 |
| | FlyingChairs | 0.863 | 0.914 | 0.126 | 0.195 |
| | Mixed | 0.896 | 0.934 | 0.117 | 0.164 |
| | FlyingThings3D | 0.894 | 0.934 | 0.090 | 0.132 |
| | KITTI | 0.714 | 0.794 | 0.031 | 0.053 |
| FlowNet2 CS | FlyingChairs | 0.841 | 0.899 | 0.137 | 0.220 |
| | FlyingThings3D | 0.847 | 0.904 | 0.075 | 0.129 |
| GMA (+P) | FlyingChairs | 0.856 | 0.912 | 0.132 | 0.212 |
| | Mixed | 0.900 | 0.936 | 0.114 | 0.179 |
| | FlyingThings3D | 0.899 | 0.936 | 0.104 | 0.171 |
| GMA | FlyingChairs | 0.864 | 0.917 | 0.131 | 0.212 |
| | Mixed | 0.900 | 0.937 | 0.090 | 0.139 |
| | FlyingThings3D + Sintel | 0.909 | 0.943 | 0.066 | 0.100 |
| | FlyingThings3D | 0.903 | 0.943 | 0.060 | 0.098 |
| | KITTI | 0.756 | 0.834 | 0.051 | 0.084 |
| GMA (P-only) | FlyingChairs | 0.846 | 0.901 | 0.128 | 0.207 |
| | KITTI | 0.766 | 0.847 | 0.092 | 0.155 |
| | FlyingThings3D | 0.903 | 0.940 | 0.083 | 0.139 |
| | Mixed | **0.912** | 0.947 | 0.077 | 0.117 |
| FlowNet2 CSS | FlyingChairs | 0.850 | 0.908 | 0.084 | 0.141 |
| | FlyingThings3D | 0.862 | 0.918 | 0.070 | 0.121 |

Table 4: Ablation study: We apply the optical flow estimators to a window of 9 frames by using the central frame as references and computing optical flow to each of the 8 other frames. The stacked optical flow fields are used as inpute for the segmentation network.

of the art OCLR model [51] in our setting. The OCLR model uses optical flow estimated by RAFT [43], which we also included in our experiments. The segmentation network however uses a U-Net architecture with Transformer bottleneck and was trained to segment multiple objects on a synthetic dataset. We use the published weights and do not retrain the model on our data.

The results in Table 5 show that the model performs very well on the original data. OCLR outperforms our motion energy based model and achieves a performance similar to the best optical flow based models considered in this work. At the same time, the model does not generalize to the corresponding random dot stimuli. These results provide further evidence that the low generalization to random dots is not due to the architecture of the segmentation network or the RGB training data, but a property of the motion estimator.

| Model | IoU (original) | IoU (random dots) |
|---|---|---|
| OCLR | 0.838 | 0.026 |
| Motion Energy Segmentation | 0.759 | 0.600 |

Table 5: Comparison of the state-of-the-art motion segmentation model OCLR, and our segmentation model based on a motion energy model.

# C   Additional details about the human subject study

## C.1   Comparison of humans and machines by example difficulty

As a measure of task difficulty, we count the number of *informative dots*. A dot is informative, if it is contained in either the target and distractor shape but not both (see Figure 7, left). Only these dots allow discriminating between the different shapes.

We fitted psychometric curves for human participants and models as a function of the number of informative dots, using the psignifit toolbox [35]. The results in Figure 7 confirm that only the motion energy model is able to match the performance of human subjects, especially for stimuli with a medium number of informative dots.

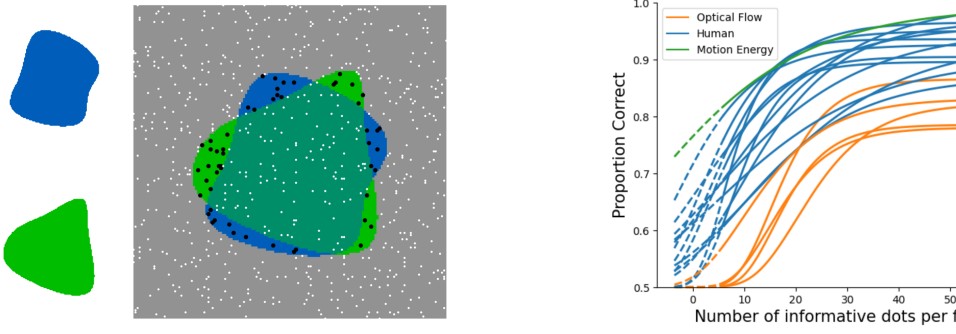

Figure 7: (*left*) As a measure of task difficulty, we count the number of informative dots that allow discriminating between the two shape alternatives. (*right*) Psychometric curves for humans, the motion energy based model and the four best optical flow models for the task as in 8.

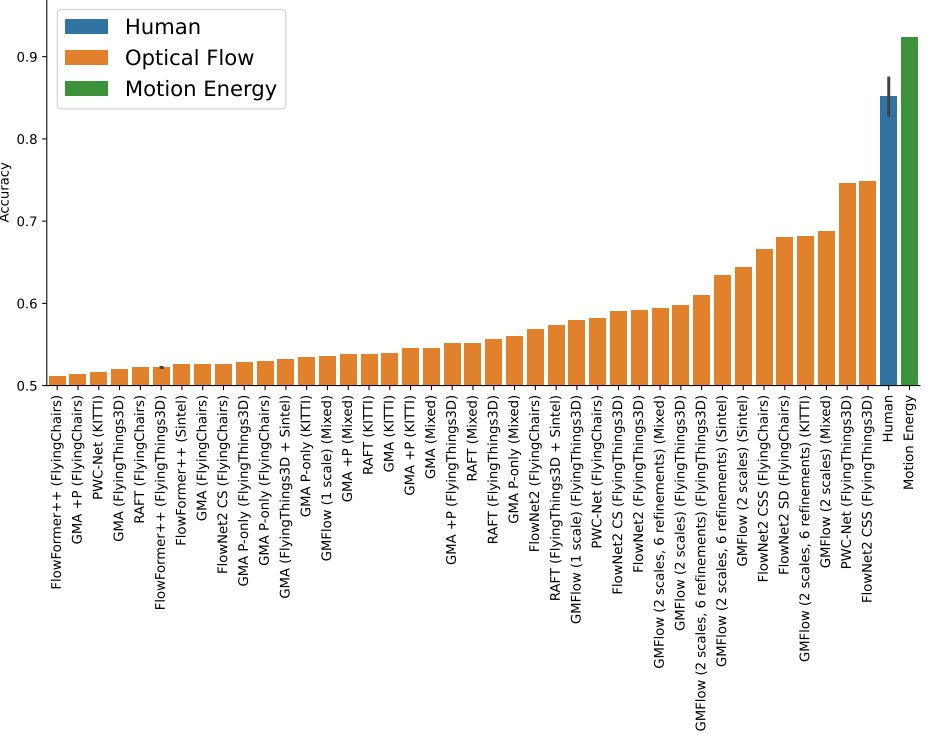

Figure 8: Comparison of the human and model performances for the random dot shape matching task.

## C.2 Screenshots of the experiment

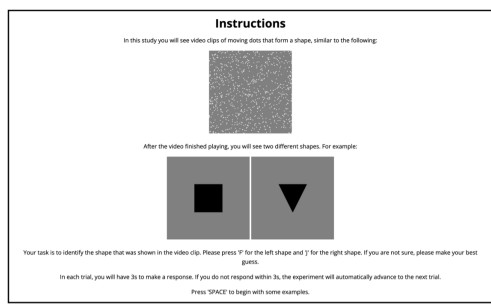
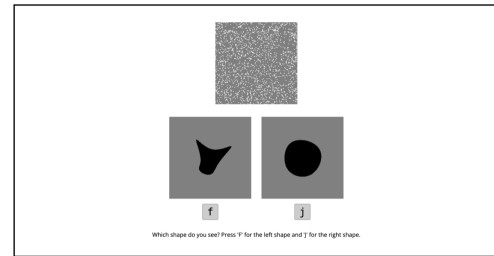

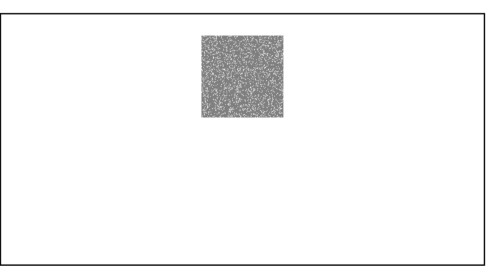
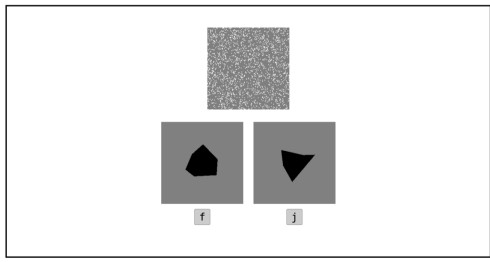

Figure 9: Screenshots from the human subject study on random dot shape identification. (*top left*) Instructions that were shown prior to the experiment. (*top right*) We showed 20 training trials during which subjects could familiarize themselves with the task. (*bottom left*) The training was followed by 500 test trials. A video with the random dot stimuli was shown first. (*bottom right*) Once the video finished playing, the two shape options were shown below.

