# OpenReview forum: "Object segmentation from common fate: Motion energy processing enables human-like zero-shot generalization to random dot stimuli"
_NeurIPS.cc/2024/Conference — NeurIPS 2024 poster_

### Official Review · Reviewer_cRYq · 2024-07-10

**Soundness:** 3
**Presentation:** 3
**Contribution:** 3
**Rating:** 6
**Confidence:** 4

**Summary:**

Authors propose a study on the generalization of motion segmentation models to random dot kinematograms, where they study various learning based (optical flow) motion estimation models vs. a classical motion energy model. Their work is the first to explore such a motion energy model for the motion segmentation task, especially with random dot scenes. They also provide human performance upper bound.

**Strengths:**

- Good contribution and novelty to study the generalization of motion segmentation methods to random dot kinematograms.
- Good experimental analysis with supportive psychophysical experiments on humans serving as an upper bound
- Comprehensive analysis of 28 Optical Flow models

**Weaknesses:**

- Missing quite relevant related work that actually utilized similar models that are not learnable but based on classical theory on these spatiotemporal oriented energy models and was evaluated on dynamic texture recognition w.r.t SOA deep learning based ones
Hadji, Isma, and Richard P. Wildes. "A spatiotemporal oriented energy network for dynamic texture recognition." Proceedings of the IEEE international conference on computer vision. 2017.

- Weak motion segmentation training data. I am wondering how the same study will conclude if the training data used included diversified and large-scale video object segmentation or motion segmentation datasets, e.g.:
              1. DAVIS: Pont-Tuset, Jordi, et al. "The 2017 davis challenge on video object segmentation." arXiv preprint arXiv:1704.00675 (2017).
              2. YouTube-VOS: Xu, Ning, et al. "Youtube-vos: A large-scale video object segmentation benchmark." arXiv preprint arXiv:1809.03327 (2018).
              3. TAO-VOS: Voigtlaender, Paul, et al. "Reducing the annotation effort for video object segmentation datasets." Proceedings of the IEEE/CVF Winter Conference on Applications of Computer Vision. 2021.

While they aren't motion segmentation datasets per se, i.e. they focus on the most appearance and motion salient objects (VOS) they still can be used to train such models as random dot kinematograms do exhibit the motion salient objects as well.

Nonetheless, still quite interesting research question, contribution, novelty and study so I am still inclined towards an accept.

**Questions:**

How is there proposed model different from SOENet cited above in the weaknesses? beyond the fact that SOENet was evaluated for dynamic texture recognition not motion segmentation.

**Limitations:**

Mainly the used training data might not be the best to conduct such study, what if their generated dataset played a role in how optical flow models generalized to such random dot scenes. A second evaluation training on large-scale VOS benchmark such as YouTube-VOS+DAVIS can help ensure the consistency of the conclusions.

---

> ### Author Rebuttal · Authors · 2024-08-06
>
> Thank you very much for your review. We’re happy you find our paper to have “good contribution and novelty” and to provide a “good experimental analysis”. In the following we would like to address your concerns.
>
> **Comparison to SOE-Net.** Thank you for making us aware of the work by Hadji and Wildes. We agree that this work should be discussed, and are happy to include a comparison to SOE-Net in our paper. Regarding the architecture, we see the following similarities and differences compared to the Simoncelli & Heeger model:
> - Both models use a separable Gaussian derivative filter, divisive normalization, a variant of quadratic nonlinearity, and Gaussian spatial pooling. Implementation details for all components differ, nevertheless we expect these components to behave very similarly.
> - Larger differences exist in the overall architecture: The Simoncelli & Heeger model is applied in parallel at different spatial scales, while spatio-temporal processing stages are applied sequentially in SOE-Net. Moreover, SOE-Net does not include a stage corresponding to MT, which is used by the Simoncelli & Heeger model to resolve ambiguities in the local motion signals.
> - The processing stages of the Simoncelli & Heeger model have been experimentally compared to neural activity. This allows us to establish a compelling link between neural mechanisms and high-level behavior. To our knowledge, it has not been established how well SOE-Net matches computations in the brain.
> - The memory and computational footprint is much higher for SOE-Net. We see two main reasons for these differences: (1) Each spatial scale in the Simoncelli & Heeger model reduces the input size by a factor of two, while the SOE-Net with default parameters uses larger feature maps (only a single reduction by a factor of 2 in the 3rd layer). (2) The layers in the Simoncelli & Heeger model use at most 28 feature maps, while SOE-Net uses 20, 400, 800 and 1600 feature maps, respectively. Taken together, these differences make it challenging to use the model for dense prediction tasks from an engineering perspective.
>
> We reimplemented SOE-Net using PyTorch, but needed to make small adjustments to reduce the memory footprint for our dense prediction setting:
> - We used a stride of 2 in the spatial pooling for each layer. The original model only decreased the resolution after the third application of the spatio-temporal model. This change also means that the model predicts the segmentation mask in only half of the input resolution. We use bilinear interpolation to predict the full resolution segmentation.
> - We replaced the global pooling in the last layer with spatial and cross-channel pooling as used in the previous layers.
>
> We use the representations of SOE-Net after each layer as multi-scale input to our coarse-to-fine segmentation network. We trained segmentation models using two variants of SOE-Net, using a square nonlinearity and the two-path square nonlinearity, respectively. As the results below show, the model performs similarly as the Simoncelli & Heeger model on the original videos. The performance gap might be explained by the reduced resolution of SOE-Net. However, SOE-Net does not generalize to random dot stimuli. We didn’t have enough time to perform further analyses, so we can only speculate on what causes this difference.
>
> | Model  	| IoU (original) | IoU (random dots) |
> | ------------- | ------------- |--------|
> | Motion Energy | 0.759 | 0.600 |
> | SOE-Net (Square) | 0.710 | 0.053 |
> | SOE-Net (Two-Path) | 0.723 | 0.052 |
>
> **Standard computer vision benchmarks.** We agree that scaling our approach to standard computer vision datasets would have been interesting and continue to explore this direction. Optical flow estimators have been heavily improved over the recent years while motion energy approaches have received hardly any attention in comparison. While we think that scaling motion energy to standard computer vision benchmarks is possible, we think the engineering effort required for that goal is beyond the scope of this work.
> As discussed in our general response, we see the main focus of our work in showing that a classical motion energy approach has more similar inductive biases compared to humans than state-of-the-art optical flow estimators. We believe that our dataset is adequate to support this claim.
>
> **Influence of training data.** To rule out a strong influence of the training data we additionally evaluated the OCLR model (Xie et al. 2022) on the random dot stimuli. To our knowledge, this is the state-of-the-art motion appearance free (i.e. purely optical flow based) motion segmentation model. This model uses optical flow predicted by RAFT using several displacements and was trained for strong performance on standard computer vision benchmarks. As the results below show, the model transfers well to our clean stimuli but fails to generalize to the random dot stimuli. This result suggests that the choice of training data is not critical.
>
> | Model  	| IoU (original) | IoU (random dots) |
> | ------------- | ------------- |--------|
> | OCLR | 0.838 | 0.026 |
> | Motion Energy | 0.759 | 0.600 |

---

> > ### Comment · Reviewer_cRYq · 2024-08-14
> >
> > After reading the author comments and other reviews, it clarified multiple issues I raised. So I am increasing to weak accept and I agree that expanding to general purpose benchmarks can be out of the current scope, the comparison to OCLR made sufficient evidence against what I was doubting initially.

---

> > > ### Author Response · Authors · 2024-08-14
> > >
> > > Thank you for your positive feedback, we're happy that we could address your concern.

---

### Official Review · Reviewer_DKrM · 2024-07-11

**Soundness:** 3
**Presentation:** 3
**Contribution:** 3
**Rating:** 6
**Confidence:** 3

**Summary:**

The manuscript investigates whether artificial neural networks perceive moving random dots similar to humans. To this end,  they propose a foreground-background segmentation framework to objectively measure several models. Two groups of models are explored: (1) driven by optical flow that finds matching points between two frames, and (2) driven by motion energy inspired by known mechanisms of the biological brain. The conclusion is that the neuroscience-inspired motion energy model matches better human perception.

**Strengths:**

1. Evaluation of artificial neural networks with random dot stimuli offers a more controlled environment to compare networks to human psychophysics.

2. A simple energy model, inspired by visual area MT in the human cortex, can open a new line of investigation to model human motion perception.

**Weaknesses:**

1. The comparison of accuracy between networks and humans is ambiguous. The performance difference between the human participants and (a) the best optical flow and (b) the motion energy model, is roughly the same but in different directions. This might suggest that both models that equally far away from human responses (the difference is that motion energy is better than humans while FlowNet2 is worse). Nevertheless, based on these results one cannot conclude that the motion energy model better captures human responses.

2. It is unclear why the online experiment has resulted in a "suspiciously low" quality outcome. Does this indicate large individual differences or suggest the experiment is demanding and can result in different interpretations by participants? If such concerns or similar arguments are valid, one must question their validity to test networks as well.

3. I would have liked to see more results from the Motion Energy model perhaps with a different set of conclusions that are illustrated in Figure 2. Perhaps some ablation studies to better investigate the role of area/component (i.e., V1/MT Linear/Blur).

**Questions:**

1. Why does neither of the entries in Table 2 match the Motion Energy results in Table 1?

2. Results of Table 2 suggest that the original weights of the proposed motion energy model from Simoncelli & Heeger [35] result in the best prediction of Random Dots, and any deviation to that deteriorates the performance rather significantly. This is rather puzzling to me. It would be interesting to read the thoughts of the authors on what kind of dataset one can obtain similar weights from Simoncelli & Heeger. Essentially, how has the human brain obtained those parameters in the first place?

**Limitations:**

The limitations of the current manuscript are well explained.

---

> ### Author Rebuttal · Authors · 2024-08-06
>
> Thank you very much for your positive review. We are happy that you think that our work “can open a new line of investigation to model human motion perception”.
>
> **Human-machine comparison.** We see our study as an ideal observer analysis and argue that humans are not expected to reach the model performance. Several participants of our study reported accidental mistakes and missed stimuli due to slipping attention during the experiment. Both are no concerns for the model. Moreover, the model computation is not affected by noise in the same way as biological systems. Finally, while the motion energy model is motivated from neuroscience the segmentation model is not and might use information more efficiently than humans. Our analysis therefore is an ideal observer analysis (Burge, Annu. Rev. Vis. Sci. 2020). This analysis shows that optical flow methods do not extract sufficient information to explain human performance while the motion energy model does. In our view this is clear evidence in favor of the motion energy approach. We agree that this point deserves more attention and will include it in the discussion of our paper.
>
> **Performance of online study participants.** Several factors probably influenced the low performance of the participants in the online study. First of all, the viewing conditions (contrast, stimulus size, brightness of the surrounding, …) were not controlled so that participants might have taken the study in a difficult setting. Moreover, we did pay subjects independently of their performance. Since the performance depends on concentration and motivation, the participants might not have entirely fulfilled their potential. As promised in the paper, we repeated the study with more people in a controlled lab setting, where participants reached consistent performances (results in the PDF). These results provide evidence against the possibility of large subjective differences.
>
> **Additional ablations.** Following your suggestions, we performed several additional ablations on the Motion Energy model: We removed the entire MT block, the MT linear layer, the pooling layers, and replaced the types of normalization and nonlinearities used in the model. The results below suggest that among the ablated components, pooling and normalization are particularly important for generalization to random dots. As discussed in our response to reviewer 8R4V, we think that the role of the pooling layers is particularly interesting regarding future improvements of the model. We will include the additional ablations in our paper.
>
> |                                   	| Original  |       	| Random Dots |       	|
> |---------------------------------------|-----------|-----------|-------------|-----------|
> | Condition                         	| IoU ↑ 	| F-Score ↑ | IoU ↑   	| F-Score ↑ |
> | RectifiedSquare -\> ReLU (MT)     	| 0.753 	| 0.838 	| **0.609**   | **0.725** |
> | Baseline                          	| 0.759 	| 0.845 	| 0.600   	| 0.718 	|
> | Square -\> ReLU (V1)              	| 0.770 	| 0.854 	| 0.536   	| 0.663 	|
> | \- MT Linear                      	| 0.768 	| 0.856 	| 0.481   	| 0.609 	|
> | \- MT                             	| 0.770 	| 0.854 	| 0.451   	| 0.583 	|
> | \- Blur (V1, MT)                  	| **0.801** | **0.872** | 0.421   	| 0.540 	|
> | ChannelNorm -\> InstanceNorm (V1, MT) | 0.592 	| 0.703 	| 0.230   	| 0.340 	|
> | \- Normalization (V1, MT)         	| 0.400 	| 0.516 	| 0.018   	| 0.018 	|
>
>
>
> **Inconsistency between tables 1 and 2.** Thank you for making us aware of this inconsistency. We used a slightly different architecture of the segmentation network for the ablation study in Table 2. We agree that this is suboptimal, and repeated the ablation study using the very same setup as for Table 1 for which we obtain consistent results.
>
>
> **Learning of motion perception in the brain.** We agree that this is a highly interesting question and are happy to extend the discussion of this result in the paper. While we cannot offer definitive truth, we have hypotheses that might inform future work in this direction:
> - We only use motion information for segmentation, while the motion based tasks performed by the brain are much more diverse (e.g., Nishida et al., Annu. Rev. Vis. Sci. 2018). Training on a more exhaustive set of tasks might be important for obtaining the parameters in the first place.
> - There might be additional inductive biases, e.g. due to the biology of the brain. In our ablation study we did not restrict the trained parameters in any way. So there is the possibility to substantially deviate from the mechanisms of the original model, e.g. beyond the spatio-temporal energy approach. Inductive biases are assumed to be central for learning in the human brain (Goyal and Bengio, 2022).

---

> > ### Comment · Reviewer_DKrM · 2024-08-13
> >
> > I thank the authors for responding to my questions and reporting complementary ablation results. I do not have any further questions.

---

> > > ### Author Response · Authors · 2024-08-14
> > >
> > > Thank you for response and taking the time to go through our rebuttal. We're happy that we could address your questions.

---

### Official Review · Reviewer_8R4V · 2024-07-18

**Soundness:** 3
**Presentation:** 3
**Contribution:** 3
**Rating:** 7
**Confidence:** 4

**Summary:**

The authors attempt to use a classical evaluation of human gestalt perception, namely shape identification in moving random dot stimuli, to evaluate both a traditional model of motion perception (motion-energy model from Simoncelli and Heeger) against state-of-the-art deep learning-based optical flow methods. They find that while SoTA methods outperform the MT motion-energy model on natural video optical flow estimation, the motion-energy model in fact outerperforms these more complex methods in correctly estimating motion patterns in random dot stimuli. The poor estimation of motion in random-dot stimuli thus correlates with poor figure-ground segmentation using the SoTA deep network approaches. Due to the fact that human performance on random-dot stimuli is far greater, this suggests that curren optical flow models fall short of human performance and might be computing motion information in a way that is misaligned with human vision. On the other hand, the fact that the motion-energy model can fill this gap suggests that this more biologically-inspired model (that fits responses of MT cells in visual cortex) is sufficient to connect cortical motion processing with some aspects of human Gestalt perception.

**Strengths:**

There are many strengths of this work and I see it as a careful and nice study of where current optical-flow methods stand relative to much simpler interpretable methods for estimating motion. The additional comparisons with human segmentation performance given random-dot stimuli are compelling and overall I think this paper provides a valuable set of experiments that can guide future work in this interdisciplinary area of human vision science and computer vision.

To be more specific, I think the work is presented quite clearly and written well. I also think that the problem framing is clear and the authors successfully verify many of their hypotheses. I encourage the authors to release all of their code and data as I think a public pytorch version of the motion energy model and this human data could be valuable to the field, and could encourage the deep learning community to consider testing motion-estimation setups with the motion energy model.

Finally,  the idea of closing the loop between motion estimation and figure-ground segmentation provides a clean test of how well models can perform in comparison to humans on this core aspect of human perception. The experiments seem relatively complete (see below for some concerns), and I find the conclusions to be relatively well supported. Overall I see this work as quite similar to the Dapello et al. 2020 paper on simulating a V1 front-end for spatial visual models, and this work provides a complement to this in the motion processing domain.

**Weaknesses:**

I have a few concerns regarding the experiments and some comparisons to prior work:

1) It seems that Yang et al. 2023 is a very related work that has not been discussed in the paper. Specifically Yang explore multiple models of optical flow and compare these to human performance in motion flow perception including natural scenes and random-dot kinematograms. While the conclusions and framing are quite different, I think this work deserves much more careful comparison. For example, in this work, the authors find that SoTA methods  (RAFT etc) do in fact correlated better with human perception than local-pooling based methods (spatial and spatio-temporal pooling). I believe the ablation of these pooling methods would be useful in this work as it could provide additional bio-inspired baselines to compare the motion-energy model to. In this vain, I also believe more work can be done to ablate aspects of the motion-energy model (as in Section 4.2) to understand what is the key component to this improved generalization. I think isolating exactly what features of the motion-energy model lead to this generalization could enhance the work greatly.

Yang, Yung-Hao, et al. "Psychophysical measurement of perceived motion flow of naturalistic scenes." Iscience 26.12 (2023).

2) While RAFT and the other methods used are strong optical flow estimators, these are still significantly older methods and perhaps comparing to a few more recent methods such as GMFlow, and FlowFormer++ (which outperform RAFT) would increase the impact of the claims

Xu, Haofei, et al. "Gmflow: Learning optical flow via global matching." Proceedings of the IEEE/CVF conference on computer vision and pattern recognition. 2022.

Shi, Xiaoyu, et al. "Flowformer++: Masked cost volume autoencoding for pretraining optical flow estimation." Proceedings of the IEEE/CVF conference on computer vision and pattern recognition. 2023.

3) I think one of the main weaknesses with the work overall is that the main results seem to suggest that the motion-energy model is much better at generalization, but significantly underperforms on the natural videos (albeit this can be improved with finetuning). I think there should be some more extended analysis of study of why this is and some more concrete proposals of how this difference can be fixed. For example, is the issue simply that the SoTA methods are very domain sensitive? For example, could you somehow finetune those methods with random-dot videos and fix the generalization gap? An analysis; however preliminary, about this would greatly increase the impact to the computer vision field in my opinion.

4) While Section 4.2 ablating finetuning aspects of the motion-energy model is quite interesting and potentially useful for the community, I believe the results are actually a bit concerning. They seem to suggest that finetuning the model does in fact overfit to the training data and then causes a large decrease in generalization to the random dot task. The authors say it is surprising that the motion-energy model (without any tuning) performs so well on the random-dot task, but I see this as perhaps a byproduct of the fact that most research on MT had been done using simple stimuli like random-dot motion and so the parameters may have initially been set to fit this data well. If this is the case, this potentially limits the scope of application of this method from the computer vision perspective as it seems like even using the motion-energy model cannot really create a "generalist model" that performs very well on multiple domains. Please correct me if I am mis-understanding, but I think the authors should comment more generally on the practical steps going forward on how this can inform creation of more human-aligned, better optical flow methods, rather than simply using this as a study to point out one flaw with current methods. At the end of the day, detecting motion in random-dot kinematograms is a great tool for understanding human and machine vision, but from a practical perspective most applications will be to natural videos or many domains. If this is the case, then there is little reason to use the motion-energy model unless you can show that there is a tangible way to use these results to improve the generality of current methods.



If the above concerns are addressed I am definitely willing to increase my score.

**Questions:**

See above weaknesses. In addition:
1) clearly based on the results in Figure 6 (appendix) random-dot shape matching is in fact significantly better for the motion-energy model than humans (while FlowNet etc. underperform). This seems to suggest that there is still a misalignment between even the motion-energy model and human perception. Do you have any thoughts on what this missing piece is and how we might be able to arrive at a better model of human perception using the motion-energy model? Perhaps because the motion-energy model doesn't really model appearance at all there is something missing? From the perspective of human vision science, I think it would be helpful to add more text discussing how this work can inform our understanding of human figure-ground segmentation and where the additional gaps may be.

2) Recent work (see below) has shown that there is a perception of opposite-direction motion in random dot kinematograms that can be verified in human vision and also seen in an application of the motion-energy model. This brings up the larger question that perhaps aligning with aspects of human vision may not be totally beneficial to computer vision systems that detect optical flow etc. Could you comment on this?

Bae, Gi-Yeul, and Steven J. Luck. "Perception of opposite-direction motion in random dot kinematograms." Visual cognition 30.4 (2022): 289-303.

**Limitations:**

The authors do address many limitations and in the text. Other than the weakenesses/questions above, I think the authors do a reasonable job of this.

---

> ### Author Rebuttal · Authors · 2024-08-06
>
> Thank you for your detailed review. We are happy that you see “many strengths” in our work.
>
> **Comparison to Yang et al. (2023).** Indeed, this work is very related since it also compares various motion estimation models to human perception. Thank you for making us aware of this paper. The focus of Yang et al. is however on naturalistic stimuli and does not consider segmentation, which limits direct comparisons. Interestingly however, Yang et al. find that FlowNet 2.0 agrees better with human perception than the more recent RAFT model which parallels the results in our work. We will extend the related work and discussion sections in our paper to include a comparison to the results of Yang et al.
> We’re not entirely certain, however, whether we correctly understood the suggested comparison to the pooling methods. The pooling baselines from Yang et al. use ground truth optical flow as input, so in our setting the performance is equal for the original and random dot videos by design—which is much closer to human perception than RAFT. Please correct us if we misunderstood your suggestion, we’re happy to include additional experiments in this direction.
>
> **Additional ablations.** Following your suggestion, we performed additional ablation studies by removing or replacing components of the motion energy. We provide detailed results in our response to reviewer DKrM. The results suggest that among the ablated components, pooling and normalization are particularly important for generalization to random dots—which confirms your suggestion that the pooling is an important component which deserves further focus. Interestingly, removing the pooling slightly improved the results on the original videos, at the expense of much worse generalization. Gaussian pooling is a relatively simple method which does not take into account object boundaries. It has been suggested that the brain uses a more sophisticated approach (e.g., Braddick et al. 1993), so we think that improving the pooling is promising for further improving the model in the future. We will include the additional results and discussion in our paper.
>
>
> **More recent optical flow models.** We included FlowFormer++ and GMFlow in our analysis and provide results in the additional PDF page. Both models support the claim of our paper: FlowFormer++ performs extremely well on clean stimuli but does not generalize to the random dot stimuli at all. GMFlow is among the best optical flow models that we tested, but like FlowNet 2.0 still falls substantially short of the motion energy model on random dot stimuli. Nevertheless, we believe this is a very interesting result since it is an exception to the trend that more recent models generalize worse which was observed so far. We are thankful for the suggestion and will include all results in our paper.
>
> **Finetuning on random dots.** We expect that training optical flow estimators on random dot stimuli would most likely improve their performance. However, as argued in the general response, the inductive biases of human motion perception allows for zero-shot generalization to random dot stimuli which recent optical flow models do not exhibit—and which is not resolved by finetuning.
> At the same time, we think it is unlikely that the differences between the motion energy model and the optical flow methods can be fully explained by manual overfitting to random dots. In particular, the model by Simoncelli & Heeger has not been optimized on random dot stimuli according to the original paper. As discussed in our response to reviewer DKrM, we think that the result that training the Simoncelli & Heeger model does not lead to a generalizable model stems from missing inductive biases or too narrow tasks. For example, we expect that including an explicit loss term for the motion predicted by the model will improve the results.
>
> **Scaling the motion energy model.** We see practical steps for improvement in combining the strengths of both approaches. Optical flow estimators exploit the rich information of deep feature representations, which are lacking in the motion energy model. At the same time, the spatio-temporal approach of the motion energy model which goes beyond the dominant two-frame paradigm turned out to be highly effective. We believe that combining these approaches is possible and will allow for both high-performing and human-aligned motion perception models. We agree that a discussion of this aspect is helpful, and will update our paper in this regard.
>
> **Differences between human performances and the motion energy model.** As argued in our response to reviewer DKrM, we see our experiment as an ideal observer analysis. Machine algorithms do not suffer from accidental mistakes, lack of concentration or noise which is inherent to biological systems. These effects could be modeled by introducing corresponding noise in the model. Moreover, as suggested in your review, we think that the lack of appearance cues in our model also could also explain some of the differences: While appearance cues are not informative for the random dot stimuli, the brain has to explicitly ignore these cues which might make this task more complex for humans.
>
> **Alignment of computer vision and perceptual models.** We agree that the goals of modeling human perception and building computer vision models of motion estimation are not completely aligned. Motion illusions, like the one you mentioned, are an important feature for models of human perception but might be considered errors in a computer vision setting. However, we think that both goals are sufficiently close to be mutually beneficial. Moreover, motion estimation is usually only a means to an end for downstream tasks. The types of illusions that humans are sensitive to don’t seem to be harmful for overall perceptual performance. So research on motion illusions might reveal “acceptable errors” for real-time and energy efficient systems that behave successfully in the real world.

---

> > ### Comment · Reviewer_8R4V · 2024-08-13
> > **Response to rebuttal**
> >
> > I thank the authors for their very clear and careful rebuttal. I believe the authors have addressed almost all of my concerns. If the authors will follow-through on the changes they say they will make in the camera-ready version then I think this will greatly strengthen the work.
> >
> > My only remaining concern is that I still think there is an over emphasis on human zero-shot generalization to random dot tasks. I think it is very hard to ever claim "zero-shot" generalization for humans because human experience is so diverse. Perhaps if you had infant or children psychophysics experiments this may be closer to a "zero-shot" experiment, but even then, there is work showing that neural selectivities in early vision may be driven for example by retinal waves even before birth. Could these kinds of retinal waves simulate something that is more akin to random dots? In this vain, I just think it would be useful to show what level of finetuning it might take for a SoTA model to generalize well to the random dot task. However, I don't believe this to be a blocker for the work as it still provides a large set of good experiments and contributions that I think will be valuable to the field. I am raising my score to a 7 as a result.

---

> > > ### Author Response · Authors · 2024-08-14
> > >
> > > Thank you for your positive feedback. We think that inductive biases are responsible for the generalization of humans to random dot stimuli, without having an explanation supported by evidence yet for what factors cause these biases. To us it seems unlikely that retinal waves before birth can fully explain the observed differences and would expect architectural biases to contribute as well.
> > >
> > > We appreciate that you think that our work "provides a large set of good experiments and contributions that I think will be valuable to the field" and agree that further exploring the origin of the inductive biases in human perception are a promising direction for future work. We are thankful for the suggestions by you and the other reviewers which yielded additional, interesting results that we are happy to integrate in our paper.

---

### Official Review · Reviewer_ezYE · 2024-07-23

**Soundness:** 2
**Presentation:** 2
**Contribution:** 1
**Rating:** 5
**Confidence:** 3

**Summary:**

The paper measures the zero-shot generalisation of motion energy (as described and estimated by Simoncelli & Heeger (1998)) when applied to the moving object segmentation task on the random-dot moving patterns. The comparison is performed with optical flow representation. Given a synthetic dataset of single-moving object scenes, either motion energy or optical flow is estimated using the Simoncelli & Heeger method or of-the-shelf pretrained network, respectively. This is fed into the downstream segmenter, which is trained in a supervised manner. This is then applied to moving random dot images. While optical flow achieves better performance in the simulated 3D object videos, the performance degrades drastically on moving dots. Motion energy modality performs worse than many recent optical flow estimators in the original setting but retains much of the performance when transferred to moving dots.

**Strengths:**

1) A multitude of optical flow estimators are tested, offering a wide coverage and interesting results that older, arguably, less data-driven approaches retained some performance on moving dots.

**Weaknesses:**

#### Presentation
1) The paper lacks a description that explains precisely what motion energy is (what form it takes and what the inputs are to estimate it). Similarly, optical flow is not described either. This is problematic as a reader unfamiliar with either of these two concepts will have a hard time making sense of the study presented.

#### Soundness

2) It is also not clear if the presented comparison is meaningful and sound. In the discussion (L290-292), the paper states that optical flow might not be optimal for humans, and that the truthfulness of flow is not critical for the task of segmentation. Why compare to it?

3) Similarly, as stated in the L302, motion energy is estimated from far more frames than flow. For moving dot patterns, the "true" flow is non-zero only for dots that have moved. This paper does not establish that this is enough to segment the shape. This is even before considering pretrained flow estimators operating outside the learned distribution. Similarly, the statement on L64 seems to misleading: optical flow and/or motion energy _are_ the motion information. It is just that optical flow does not work on moving dots whereas motion energy seems to still extract signal (perhaps due to many-frame observation).

4) Does running optical flow on dense moving dot patterns even make sense? It is worth considering whether obtaining "true" optical flow between two frames is ill posed in this situation. Over a larger number of frames, it perhaps is the simplest explanation of the motion consistent with the overall trend. Given just two frames with indistinguishable dots, there could be many possible "simple" motions that rearrange the dots. It is perhaps too much to expect that the appropriate optical flow could be estimated with only two frames, and that the estimate will have enough information to describe the shape (which is essentially the overall trend of motion in these simple cases).

5) There seems to be a premise in the paper that generalising to the moving dot patterns is desirable and important. This is not motivated. Is it even true? One would usually expect to apply computer vision systems more to the real-world setting than to moving dot setting. Based on the results in Table 1, recent optical flow estimators are preferable in this situation. Based on further study in Table 2, optimising the motion energy model weights to perform better in a more "real" setting reduces performance on moving dots. Why is the generalisation to the moving dot settings relevant?

#### Limited Contribution
7) While the presented study is novel, its usefulness is not argued or clear. It is not obvious how the findings could be used, which essentially are the measurements performed. The motion energy model is taken from prior works, does not improve with training in moving dot patterns and under-performs in the real setting. Overall, the advancement of the knowledge presented in the paper is minimal.

**Questions:**

The rebuttal could concentrate on answering the critical question presented above in the weaknesses section.

#### Additional Remarks
 - The size of the datasets and evaluation protocol for "Original" and "Moving dot" settings are not described. As the datasets are new, details are required to reproduce and verify the results.
 - How are the inputs to the segmentation network processed? Are the motion energy/optical flow normalised and/or transformed in some way?
 - On L78, it is said that the study of moving object segmentation is conducted, while the focus of the study is really the moving dot segmentation setting.
 - The presented motion energy model outperforms all human subjects in the study. While possible, this is quite strange and could indicate that the small sample size and/or task set up are inappropriate.

**Limitations:**

The key limitations are addressed.

---

> ### Author Rebuttal · Authors · 2024-08-06
>
> We thank you for your thoughtful review. As argued in our general response, we see the main contribution of our work not to computer vision but to cognitive neuroscience; by establishing a compelling link between a biologically motivated model and high-level inductive biases of human perception.
>
> **Defining motion energy and optical flow.** Thank you for the suggestion. We will update our paper to review the concepts of motion energy and feature-matching based optical flow. In a nutshell, we’re comparing two approaches for motion estimation: Matching features between two frames and using spatio-temporally oriented filters (3d convolution). This is also the difference we ment in our statement in L.64, which we will clarify accordingly.
>
> **Comparison to optical flow estimators.** Deep learning based optical flow estimators have achieved remarkable accuracy during recent years. Due to this success, we find it natural to consider this class of models for our goal of building a model for motion perception in humans.
> While the goals of computer vision and modeling human vision differ in some aspects (for example regarding motion illusions) we believe that both lines of work are sufficiently close to be mutually beneficial.
>
> **Number of input frames.** The different numbers of frames used by motion energy and optical flow methods is a valid concern in our view. We therefore conducted an additional ablation study in which we applied optical flow using the same number of frames as motion energy. As is common for motion segmentation, we are using multiple gaps of [-4, -3, …, +3, +4] to the central frame and using the concatenated optical flow as input to the segmentation network. The results in the additional PDF show that optical flow models are unable to achieve the same generalization to random dot stimuli as motion energy even when using the same number of frames.
>
> **Relevance of random dot stimuli.** We agree that the “true” optical flow for random dots is ambiguous and believe that this makes random dot stimuli particularly interesting. The consistent perception of humans demonstrates that the sparse information of these stimuli is enough to perceive shapes, which reflects inductive biases of the underlying motion perception system. While the physically correct prediction might not be well defined, a good model for human perception clearly should reflect these biases. Moreover, random dot stimuli are well established in the literature on human motion perception. If models handle these stimuli similar to humans, this opens many opportunities for comparison with previous data.
>
> **Relevance for computer vision.** The main goal of our work is to improve models for human motion perception and interpretation. While clean videos are more relevant for practical applications of computer vision, random dot stimuli are a highly important stimulus class in human vision. Nevertheless, we think that this study also provides some directions for building better computer vision models. Given that the motion energy model has several orders of magnitude fewer parameters compared to the optical flow models (<2.5K), the model performs remarkably effectively. We hypothesize that the spatiotemporal filtering efficiently aggregates information from multiple frames, which is an operation that could well be included with computer vision models and help to identify more efficient architectures.
>
> **Contribution.** We argue that our study makes several valuable contributions for modeling human perception and fits the “neuroscience and cognitive science” subject area of NeurIPS very well.
> - The motion energy model was not developed as an image-computable model for end-to-end machine learning, but to explain the behavior of individual neurons. We show that this model can nevertheless be used as a DNN component for the computer vision task of motion segmentation.
> - Our knowledge of how humans organize motion patterns into a higher-level scene representation is still limited (e.g., Nishida et al. 2018). Our work makes a valuable contribution to this end by showing how a well understood, low-level mechanistic model of neural activity leads to inductive biases that agree with human scene perception.
> - Random dot stimuli have a long tradition in vision science. Nevertheless we are the first to measure segmentation of random dot stimuli in humans. We believe that this experimental data is a valuable contribution to the psychophysics of human perception.
> The other reviewers seem to agree with us on these points and write that “this paper provides a valuable set of experiments that can guide future work in this interdisciplinary area” (834V), denote our work as a “Good contribution and novelty” (crYq) and posit that it “can open a new line of investigation to model human motion perception” (DKrM).
>
> **Additional Remarks.** Thank you for your feedback, we’re happy to improve the clarity of our paper in these regards.
> - Our new dataset consist of 901 training and 100 test videos, each having 90 frames at 30Hz. We use distinct backgrounds and foreground objects for both subsets. Models predict a binary segmentation with distinct foreground and background channels. So the IoU and F-Score metrics are computed as in for semantic segmentation, but using 2 classes.
> - Motion energy and optical flow are not normalized or transformed.
> - We argue that moving object segmentation is exactly the task which is targeted in our work, but are happy to clarify that we’re focussing on the generalization to random dot stimuli.
> - Compared to humans, our model does not suffer from accidental mistakes and slipping attention. Please also see our response to reviewer DKrM where we discussed this aspect in greater detail.

---

> > ### Comment · Reviewer_ezYE · 2024-08-13
> >
> > I thank the authors for their reply and for improving the clarity of their work. I have updated my rating to reflect that.
> >
> > I still have my reservations about the meaningfulness of comparing to optical flow:
> > > Due to this success, we find it natural to consider this class of models for our goal of building a model for motion perception in humans
> >
> > Yet, L291-292 states that "estimating the physically correct optical flow might not even be optimal for humans in terms of processing efficiency." The study seems to largely confirm that optical flow is not great for motion perception in humans.
> >
> > > we are using multiple gaps of [-4, -3, …, +3, +4] to the central frame and using the concatenated optical flow as input to the segmentation network.
> >
> > I appreciate the authors conducting additional experiments. Optical flow predictors only work with a pair of frames to assess motion information rather than a whole stack of them, so stacking multiple wrong estimates would not create a good long-term one. A closer comparison, perhaps in future work, could be to consider point-tracking algorithms like PIPS++, CoTracker, or BootsTAP, which reason over multiple frames to estimate coordinate change in all of them.
> >
> > However, I am still somewhat sceptical that would work for examples of random dot stimuli images. Flow and tracking models all search for matching features trained to represent varying textures and semantic wholes, while the random stimuli data specifically removes low-level visual descriptors, which these systems rely on. While the study highlights the expected "lack" in flow models, the motion energy does not seem to be the answer either.

---

> > > ### Author Response · Authors · 2024-08-14
> > >
> > > Thank you for your positive feedback. The observed differences between feature-matching based optical flow and motion energy are indeed plausible given the respective motion estimation mechanisms, but has not been empirically investigated before. Therefore we see our work as a substantial contribution to computational neuroscience by clearly highlighting these differences and their consequences for similarities to human perception. Exploring how to resolve these differences with a model that combines the respective strengths of both approaches is a promising direction for future work, and we thank the reviewer for their suggestions in this direction.

---

### Author Rebuttal · Authors · 2024-08-06

We thank all reviewers for their thoughtful reviews and are happy that the reviewers see “many strengths” (8R4V) and that our work “can open a new line of investigation to model human motion perception” (DKrM). The reviews also contain concerns and valuable suggestions that we are happy to address. In this general reply we will respond to concerns raised by several reviewers, and we respond to more detailed points in the responses to the respective reviews.

The primary goal of our work is finding good models for *human* motion perception. We focus on the fact that humans generalize in a zero-shot manner to random dot stimuli. Since generalization is zero-shot and since motion is not uniquely defined for these sparse stimuli, this setting reveals distinct inductive biases that underlie human motion perception. A good model of human motion perception should exhibit the same inductive biases. In our work, we show that a broad range of high performing optical flow estimation networks are substantially outperformed by a simple, classical motion energy model in this regard. Despite inferior performance on clean data, the motion energy model therefore captures an important aspect of human vision that is missing in state-of-the-art optical flow models. We believe that this is a valuable result towards understanding human motion perception.

Improving computer vision approaches to motion estimation on standard benchmarks is only a secondary goal of this work. Optical flow estimators have been steadily improved during the last decades while the motion energy model was published almost 30 years ago and was designed for explaining responses of individual neurons, not as an image-computable motion estimator. Inferior results on clean data and standard computer vision benchmarks is therefore not surprising in our view. Nevertheless, we think the relatively simple motion energy model with several orders of magnitude fewer parameters (<2.5K) is remarkably effective and provides valuable insights also for improving computer vision. Recent optical flow methods focus on matching deep feature representations between two frames while motion energy models are based on spatiotemporal filtering using a longer temporal context. We hypothesize that combining both approaches enables models that both perform well on standard benchmarks and behave similar to human perception. While we are convinced of the feasibility, we think that scaling classical motion energy models to performance levels that are competitive with state-of-the-art approaches on computer vision benchmarks is beyond the scope of this work.

We’re happy to more clearly describe the motivation and contribution of our work and to extend the discussion to provide an outlook on how to scale motion energy models on practical computer vision tasks, as outlined above. Moreover, we have conducted several additional ablation studies and comparisons as suggested by the reviewers, which further support the central claims of our paper:

- We additionally included the more recent FlowFormer++ and GMFlow models in our work. The results confirm the claims made in our submission.
- We additionally included SOE-Net as a motion representation. This model is based on similar components as the Simoncelli & Heeger model, but applies the spatio-temporal filter repeatedly in order to obtain a feature hierarchy. While reaching comparable performance on the original videos, this model does not generalize to random dots. We hypothesize that the repeated application of the spatio-temporal filter and the cross-channel pooling substantially changes the inductive bias of the model.
- (*) We performed an additional ablation study in which we removed components (MT / MT Linear / Pooling / Normalization) or replaced them with other variants (InstanceNorm, ReLU). The results suggest that the normalization and pooling operations are particularly important for generalization.
- (*) We additionally evaluated the optical flow models using 9 frames as input, matching the input length of the motion energy model. This change did not improve generalization to random dots.
- (*) We evaluated the OCLR model (Xie et al. 2022) on our data. This is a state of the art motion segmentation model based on RAFT and optimized for performance on computer vision benchmarks. Confirming the claims in our paper, this model performs well on the original videos but does not transfer to random dots.
- We repeated the human subject study in a controlled vision lab using more participants (13). We obtained similar results as in our preliminary, less controlled study.

We share more detailed results in the attached PDF, and in the responses to reviewers DKrM and cRYq. We will include all additional results in the paper. To match the space constraints we plan to move results marked with (*) to the supplement. Please let us know if you find a different organization to be more adequate.

---

### Decision · Program_Chairs · 2024-09-25

**Decision:**

Accept (poster)

**Comment:**

Reviews were initially balanced about acceptance. Though the discussion period has resolved several concerns and all reviewers now agree that the paper is a valuable contribution.

The reviewers reported several strengths :
- "A multitude of optical flow estimators are tested"
- " novelty to study the generalization of motion segmentation methods to random dot kinematograms"
- "Good experimental analysis with supportive psychophysical experiments on humans"
- "Evaluation of artificial neural networks with random dot stimuli offers a more controlled environment to compare networks to human psychophysics."
- "the authors successfully verify many of their hypotheses"
- "provides a clean test of how well models can perform in comparison to humans on this core aspect of human perception"

The authors are expected to update their paper for the camera-ready version by including :
- comparison to the results of Yang et al,
- the extra ablation studies,
- the results with FlowFormer++ and GMFlow,
- other reviewers requests and promises that were made during the rebuttal that might have missed...